# The association between osteopontin and tuberculosis: A systematic review and meta-analysis

**Dongguang Wang, Xiang Tong, Lian Wang, Shijie Zhang, Jizhen Huang, Li Zhang, Hong Fan**[ORCID]*

Department of Respiratory and Critical Care Medicine, West China Hospital/West China School of Medicine, Sichuan University, Chengdu, Sichuan, China

* fanhongfan@qq.com

**Data Availability Statement:** All relevant data are within the paper and its Supporting information files.

## Abstract

### Objective

We examined the data reported in the studies for comparison of osteopontin (OPN) levels in tuberculosis and healthy participants, and to discuss whether OPN could be extended to disease diagnosis, severity assessment and therapeutic effect monitering.

### Methods

A systematic literature search was conducted in PubMed, EMBASE, Scopus, the Cochrane Library, Web of Science, the China National Knowledge Infrastructure (CNKI) and WanFang databases. The pooled risk estimates were shown in standardized mean difference (SMD) with 95% confidence interval (CI) for OPN levels. The random effect model was used according to the test of heterogeneity among studies. Subgroup analyses and meta-regression models were performed to identify the possible sources of heterogeneity.

### Results

17 retrospective studies with 933 tuberculosis participants and 786 healthy controls were finally included in this article. In the primary meta-analysis, higher serum/plasma OPN levels were found in tuberculosis patients (SMD = 2.58, 95%CI = 2.09~3.08, $P<0.001$). Besides, pooled results from positive acid-fast bacilli (AFB) staining and imaging-severe tuberculosis group demonstrated higher OPN concentrations (SMD = 0.90, 95%CI = 0.58~1.21, $P<0.001$; SMD = 1.11, 95%CI = 0.90~1.33, $P<0.001$; respectively), and OPN levels decreased after two months of standard anti-tuberculosis therapy (SMD = 2.10, 95%CI = 1.36~2.85, $P<0.001$).

### Conclusions

Elevated serum/plasma OPN levels may be associated with an increased risk of tuberculosis, while further well-designed studies are needed. Moreover, OPN could be considered as a potential biomarker for tuberculosis surveillance and severity assessment.

**Funding:** HF received the awards. This original study was supported by National Key R&D Program of China (2017YFC1309703) and 1•3•5 project for disciplines of excellence–Clinical Research Incubation Project, West China Hospital, Sichuan University (2019HXFH008). The funders of this research did not contribute to the study design development, analysis, interpretation of data and in writing the manuscript.

**Competing interests:** The authors have declared that no competing interests exist.

## Introduction

Tuberculosis (TB) is an infection by *Mycobacterium Tuberculosis* (MTB), remaining the leading cause of death from infectious diseases in adults globally. According to the WHO global report, there were an estimated 1.2 million deaths among HIV-negative TB patients around the world in 2018 [1]. In many settings, the relative absence of simple, reliable and dynamic monitoring indicators is still a major threat to TB controls, although multiple developments have been made in the field of tuberculosis biomarkers. Currently, numerous promising candidates are identified for risk of TB infection, severity of illness and response to anti-tuberculous treatment, most of whom are host-derived and not available on the market [2].

Osteopontin (OPN), also known as secretory phosphoprotein 1, bone sialoprotein 1, etc., is a highly phosphorylated glycoprotein. It acts as an extracellular matrix protein and immune modulator existing in a large number of tissues such as epithelium, salivary and mammary glands, kidney, brain, bone and teeth [3], which could be secreted into all body fluids. Excessive and deregulated OPN expression links to a variety of physiological and pathological processes including cell adhesion and migration, angiogenesis, host immune response, wound healing, neurodevelopment and tumor metastasis [4]. In respiratory diseases, OPN is among the most abundantly expressed proteins, regulating aspects of airway remodeling, pulmonary fibrosis, and malignancy [5, 6]. Recently, OPN has been found participating in the process of asthma, chronic obstructive pulmonary disease (COPD), pulmonary hypertension (PH) and lung cancer, while the potential in tuberculosis remains controversial. OPN may participate in granuloma formation of TB and sarcoidosis, and reportedly plays an essential role in host resistance against TB and LTBI [7, 8]. However, results from Gerritje et al. suggested an inconsequential role of OPN upon the protective immunity to MTB infection [9]. Here, we conducted this systematic review and meta-analysis to comprehensively understand the function of OPN in tuberculosis.

## Materials and methods

### Search strategy

This study was carried out following the Preferred Reporting Items for Systematic Reviews and Meta-Analyses (PRISMA) guidelines. We performed the systematic review and meta-analysis using PubMed, EMBASE, Scopus, the Cochrane Library, Web of Science, the China National Knowledge Infrastructure (CNKI) and WanFang databases to identify studies up to April 30, 2020. We used the searching terms ("tuberculosis" OR "Koch disease" OR "mycobacterium tuberculosis infection") AND ("osteopontin" OR "sialoprotein 1" OR "secreted phosphoprotein 1" OR "uropontin" OR "SPP 1" OR "OPN"). The retrieved studies were restricted to English or Chinese.

### Study selection and exclusion

The included studies met the following conditions: (1) the study used quantitative laboratory-based assays to measure the levels of OPN in tuberculosis patients and healthy controls; (2) data available, the concentration of OPN was reported as mean and standard deviation (SD) or could be converted into this expression format; (3) study subjects: human beings; (4) initial treating tuberculosis. In this study, we only used data published in English and Chinese. Abstracts, conference papers, repeated publications or literature with too little information to extract details were excluded.

## Data extraction and quality assessment

The literatures were screened for relevance by reading titles and abstracts, and then read for full texts by two independent authors (DGW and LW), and the third author (XT) was consulted to resolve disagreements. The information extracted included author, country, publication year, participant characteristics, immune status, diagnosis, diagnosing method, specimen type, OPN concentrations and detection method.

If a study only provided medians and ranges (or interquartile ranges [IQR]), we converted the data to approximately the mean and standard deviation following the validated method proposed by Wan et al. [10].

The quality of nonrandomized studies were evaluated using the Newcastle-Ottawa Scale (NOS). Briefly, this scale assigns four, two and three points for patient selection, comparability and exposure evaluation, respectively. Nine-point means the best quality, while zero-point means the poorest quality [11].

## Statistical analysis

For all analyses, two-side $P<0.05$ were considered statistically significant. Chi-squared and $I^2$ tests were used to assess the heterogeneity of the clinical trial results. When the Chi-squared test P-value was$<0.1$ and the $I^2$ test had a value$>50\%$, it suggested a statistically significant heterogeneity and the random-effects model was used to directly compare the level of OPN in cases with that in controls. Otherwise, a fixed-effects model was selected. Additionally, the meta-regression and sensitivity analysis were used to estimate the sources of heterogeneity, and the visual inspection of asymmetry in funnel plots was used to assess publication bias and the Begg's and Egger's tests were used to further detect publication bias. If there exists publication bias, we conducted a trim-fill adjusted analysis and recalculated the effect size (ES) to remove the asymmetry of the funnel plot [12]. All statistical analyses were performed using the RevMan 5.2 and STATA 12.0.

## Results

### Characteristics and quality of published studies

349 records were identified from the initial search strategy. After screening titles and abstracts, 258 full-text articles were further assessed for eligibility, and 17 studies were finally included (Fig 1), in which 933 tuberculosis patients (including 51 spinal tuberculosis and 882 pulmonary tuberculosis individuals) and 786 healthy controls were contained. All studies focused on adult populations and reported serum/plasma OPN concentrations in tuberculosis patients and healthy controls. Among the included studies, 2 reported HIV uninfected patients [13, 14] and 14 in non-immunocompromised hosts (no HIV infection, systematic autoimmune disorders and collagen diseases, glucocorticoid or immunosuppressant using, and malignancies) [15–28], in the remaining study the immune status was not described [29]. The quality of studies was evaluated by NOS, and all of 17 studies achieved five or more stars. Main characteristics of each study included were summarized in Table 1.

### Quantitative results (meta-analysis)

Overall, the pooled data demonstrated that serum/plasma concentrations of OPN in tuberculosis patients were higher than those in healthy individuals by the random-effect model (SMD = 2.58, 95%CI = 2.09~3.08, $P<0.001$, $I^2$ = 93%) (Table 2, Fig 2). To explore the potential causes of heterogeneity, we conducted a subgroup analysis by sample type (plasma and serum), and the result showed no change to heterogeneity. Furthermore, we performed a

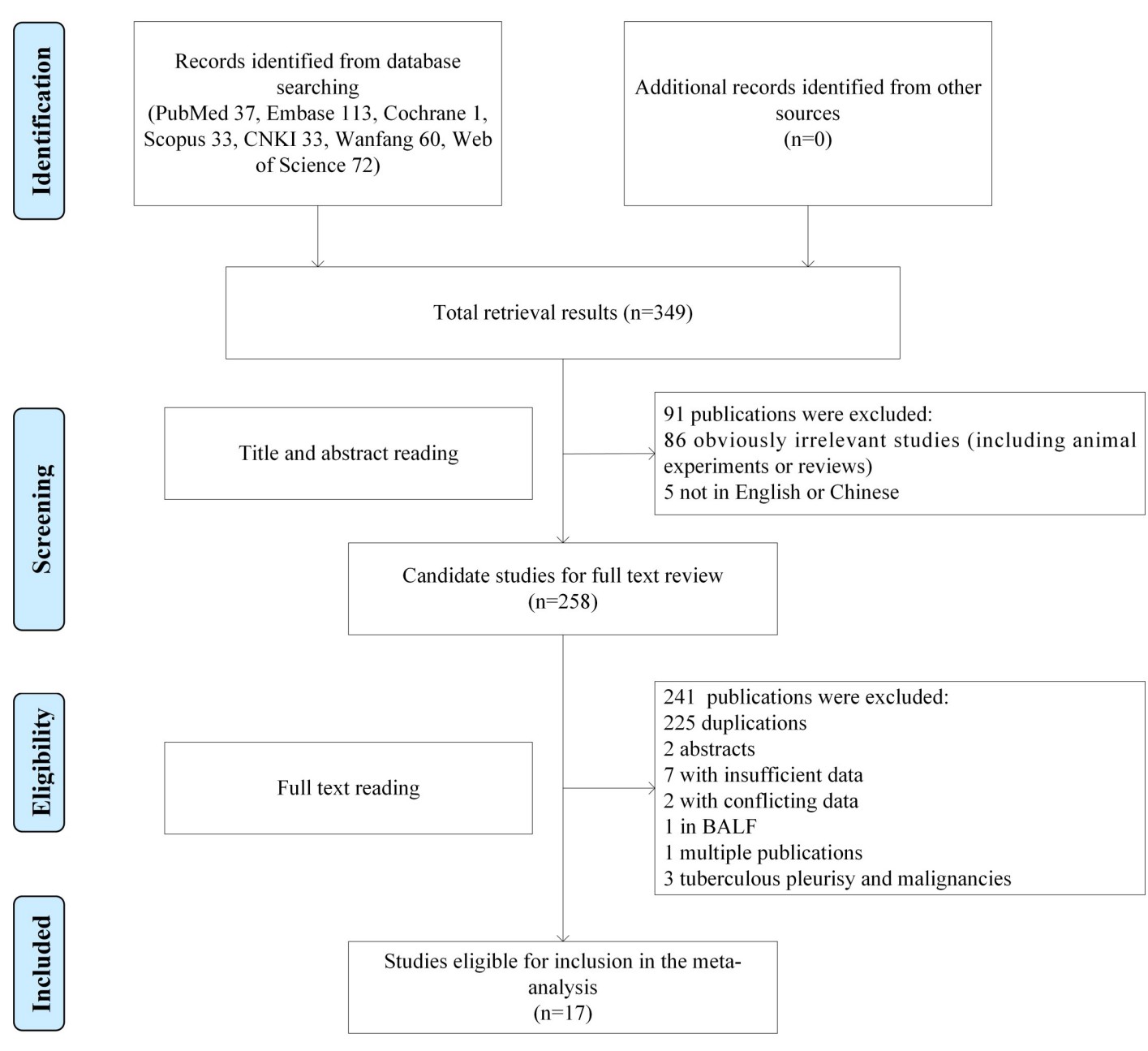

**Fig 1. The flow diagram of included and exclude studies.**

meta-regression analysis using the following covariates: country, sample size, gender, average age, tuberculosis type, and immune status, while the results didn't indicate any possible cause contributing to the heterogeneity ($P$ = 0.084, 0.981, 0.978, 0.968, 0.893, 0.181, respectively). Besides, the sensitivity analysis showed that the pooled SMD were not materially altered, suggesting the stability of the result.

Besides, 6 studies [16, 20, 23–26] explored the relation between OPN concentrations and microscopic observation of AFB staining in sputum, and the pooled data suggested that positive AFB staining was significantly associated with higher OPN levels, with the pooled SMD of

**Table 1. Main characteristics of the eligible studies.**

| Study | Patients' country | Published year | Size Case/control | Gender(male) Case/control | Age (years) Case/control | ICH | Diagnosis | Specimen type | Diagnosing method | NOS score |
|---|---|---|---|---|---|---|---|---|---|---|
| Shiratori B | Japan | 2014 | 37/30 | 23/9 | 39.5±9.88/ 34.5±9.07 | HIV (-) | PTB | Plasma | AFB staining, clinical symptoms and chest radiographs | 7 |
| Shiratori B | Japan | 2017 | 36/19 | 28/12 | 56.0±15.84/ 38.5±13.00 | HIV (-) | PTB | Plasma | Culture | 5 |
| Inomata SI | Japan | 2003 | 47/25 | NA/16 | NA/23-62 | None | PTB | Plasma | Culture or histologically proven | 5 |
| Zhang YT | China | 2018 | 48/53 | 32/36 | 36.1±9.7/35.3 ±10.2 | None | PTB | Serum | NR | 5 |
| Wu TF | China | 2017 | 51/63 | 21/29 | 41.85±11.55/ 39.73±10.06 | None | Spinal tuberculosis | Serum | Histopathology | 7 |
| Ding YL | China | 2017 | 20/20 | 13/NA | 55.0±19.8/ 45.0±21.9 | None | PTB | Plasma | AFB staining | 8 |
| Hao XL | China | 2016 | 43/40 | 23/20 | 40.5±6.5/38.9 ±5.8 | None | Smear-negative PTB | Serum | Histopathology | 8 |
| Cui JX | China | 2014 | 80/100 | 54/67 | 44.6±12.3/ 41.9±10.6 | NR | PTB | Serum | NR | 7 |
| Guo SX | China | 2013 | 42/36 | 31/28 | 34.6±10.1/ 31.2±8.6 | None | PTB | Serum | NR | 7 |
| Gan B | China | 2009 | 44/20 | 28/14 | 39.5±13.67/ 30.5±9.10 | None | TB | Plasma | NR | 6 |
| Du H | China | 2020 | 50/20 | 37/14 | 52.6±10.7/ 49.8±11.2 | None | PTB | Serum | AFB staining | 8 |
| Wang Q | China | 2019 | 92/50 | 52/31 | 43.68±9.84/ 43.19±9.76 | None | PTB | Serum | NR | 7 |
| Wu YX | China | 2019 | 90/90 | 56/51 | 49.6±17.2/ 47.6±18.2 | None | PTB | Serum | PPD skin test, AFB staining, chest radiographs, etc. | 7 |
| Feng PL | China | 2017 | 55/60 | 38/41 | 35.7±9.5/36.1 ±8.6 | None | PTB | Serum | NR | 5 |
| Yuan Y | China | 2016 | 52/45 | 36/35 | 35.5±11.2/ 32.2±9.7 | None | PTB | Serum | NR | 5 |
| Qiao YF | China | 2013 | 62/35 | 44/22 | 41.9±10.5/ 41.2±10.7 | None | PTB | Serum | NR | 5 |
| Sun XX | China | 2016 | 84/80 | 65/41 | 57.72±16.43/ 56.95±19.37 | None | PTB | Serum | Histopathology | 6 |

ICH, immune-compromised host; HIV, human immunodeficiency virus; NA, not available; NR, not reported; PTB, pulmonary tuberculosis; TB, tuberculosis; AFB staining, acid-fast bacilli staining; PPD, purified protein derivative; NOS, Newcastle-Ottawa Scale.

0.90 (95%CI = 0.58~1.21, $P<0.001$, $I^2$ = 51%) (Fig 3). In addition, according to the imaging findings, PTB could be further divided into two types: severe PTB (including military tuberculosis and cavitary tuberculosis) and non-severe PTB (including infiltrative tuberculosis and tuberculous pleurisy), and 7 studies [16, 20, 21, 24–26, 29] compared the OPN levels in PTB patients with severe imaging presentations to those with non-severe findings, and the result showed that higher OPN levels were related to severe tuberculosis (SMD = 1.11, 95% CI = 0.90~1.33, $P<0.001$, $I^2$ = 47%) (Fig 4). Moreover, the pooled result from 4 studies [19–22] with follow-up data revealed a significant decrease in OPN concentrations after two months of intensive therapy with anti-tuberculosis drugs (SMD = 2.10, 95%CI = 1.36~2.85, $P<0.001$, $I^2$ = 85%) (Fig 5).

**Table 2. Main results of the eligible studies.**

| Study | Diagnosis | Specimen type | OPN variants | Cases | | | Controls | | | Unit | Detection method |
|---|---|---|---|---|---|---|---|---|---|---|---|
| | | | | Mean | SD | N | Mean | SD | N | | |
| Shiratori B 2014 | PTB | Plasma | NR | 150.73 | 53.43 | 37 | 74.23 | 19.34 | 30 | ng/mL | ELISA |
| Shiratori B 2017 | PTB | Plasma | Full-length and cleaved OPN | 59.46 | 37.70 | 36 | 40.05 | 27.72 | 19 | ng/mL | ELISA |
| Inomata SI 2003 | PTB | Plasma | Full-length OPN | 433.0 | 259.0 | 47 | 170.0 | 65.9 | 25 | ng/mL | ELISA |
| Zhang YT 2018 | PTB | Serum | NR | 657.5 | 315.6 | 48 | 170.3 | 114.2 | 53 | ng/L | ELISA |
| Wu TF 2017 | Spinal tuberculosis | Serum | NR | 538.94 | 258.41 | 51 | 143.12 | 78.31 | 63 | ng/mL | ELISA |
| Ding YL 2017 | PTB | Plasma | NR | 563.2 | 123.4 | 20 | 40.2 | 11.2 | 20 | ng/mL | ELISA |
| Hao XL 2016 | Smear-negative PTB | Serum | NR | 712.62 | 335.75 | 43 | 185.34 | 120.23 | 40 | pg/mL | ELISA |
| Cui JX 2014 | PTB | Serum | NR | 588.42 | 271.67 | 80 | 169.12 | 97.43 | 100 | ng/mL | ELISA |
| Guo SX 2013 | PTB | Serum | NR | 683.21 | 321.76 | 42 | 168.46 | 117.31 | 36 | ng/mL | ELISA |
| Gan B 2009 | TB | Plasma | NR | 160.54 | 32.42 | 44 | 76.18 | 20.69 | 20 | ng/mL | ELISA |
| Du H 2020 | PTB | Serum | NR | 564.52 | 126.51 | 50 | 40.26 | 11.72 | 20 | ng/mL | ELISA |
| Wang Q 2019 | PTB | Serum | NR | 663.45 | 94.20 | 92 | 384.14 | 37.15 | 50 | ng/mL | ELISA |
| Wu YX 2019 | PTB | Serum | NR | 663.9 | 282.6 | 90 | 356.4 | 130.2 | 90 | pg/mL | ELISA |
| Feng PL 2017 | PTB | Serum | NR | 691.54 | 102.72 | 55 | 171.94 | 36.83 | 60 | ng/mL | ELISA |
| Yuan Y 2016 | PTB | Serum | NR | 683.2 | 321.8 | 52 | 168.5 | 117.3 | 45 | ng/mL | ELISA |
| Qiao YF 2013 | PTB | Serum | NR | 652.4 | 270.8 | 62 | 155.6 | 107.2 | 35 | ng/mL | ELISA |
| Sun XX 2016 | PTB | Serum | NR | 652.31 | 264.74 | 84 | 152.93 | 113.26 | 80 | ng/L | ELISA |

PTB, pulmonary tuberculosis; TB, tuberculosis; SD, standard deviation; NR, not reported; ELISA, enzyme-linked immunosorbent assay.

## Publication bias

A visual inspection of funnel plot demonstrated asymmetry, and this was further confirmed by Egger's test with $P = 0.004$, although the Begg's test didn't indicate a statistical significance ($P = 0.174$). Based on that, we performed a trim-and-fill analysis and the result showed that 3 studies might be missing. As shown in Fig 6, the circle represented the studies initially included in this meta-analysis, the square represented the 3 studies added by trim-and-fill analysis, and the black funnel represented the adjusted Begg's test. When these studies were added, the adjusted SMD was 2.987 (95%CI = 2.366~3.609, $P<0.001$), indicating that there was still a statistically significant association between serum/plasma OPN concentrations and tuberculosis.

## Discussion

Tuberculosis is a contagious infectious disease caused mainly by *Mycobacterium Tuberculosis*. In humans, the adaptive immune responses to *M. Tuberculosis* primarily hinge on antigen specific CD4$^+$ T cell response. Several studies have revealed the critical role of T cell immunity in the control of tuberculous infection, and defects in T cytokine production, particularly interferon-γ (IFN-γ), are genetically responsible for the development of human tuberculosis disease [30]. Osteopontin (OPN) is a highly negatively charged, arginine-glycine-aspartate (RGD)-containing and O-glycosylated phosphoprotein with little or no detectable tertiary structure by nuclear magnetic resonance (NMR) spectroscopy, encoded by a single gene clustered on chromosome 4 in human beings and produced by several types of cells such as osteoclasts, endothelial cells, epithelial cells and immune cells [31, 32]. It contains several cell interacting domains, with accumulating evidence revealing its stimulation to signal transduction pathways via RGD-dependent ($\alpha_V\beta_1$, $\alpha_V\beta_3$ or $\alpha_V\beta_5$) and RGD-independent ($\alpha_4\beta_1$, $\alpha_5\beta_1$, $\alpha_8\beta_1$ or $\alpha_9\beta_1$)

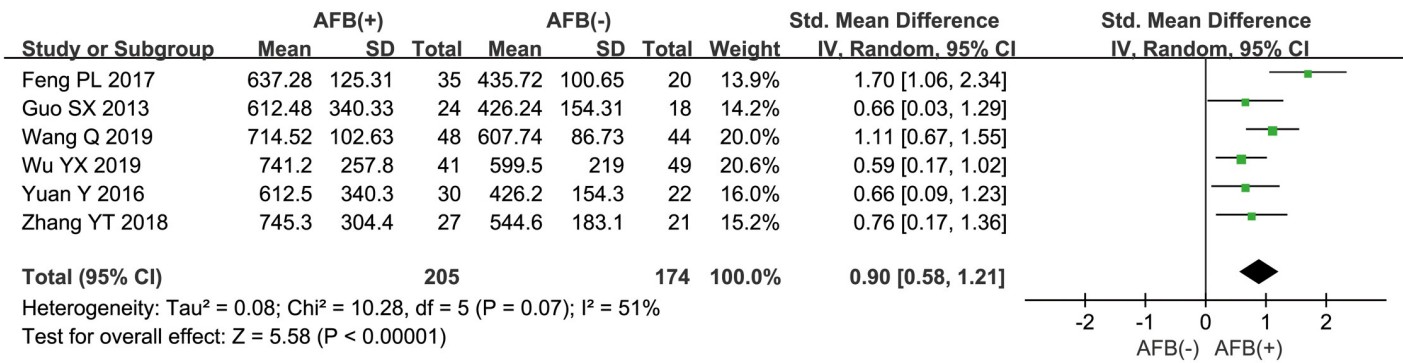

| Study or Subgroup | Tuberculosis Patients | | | Healthy Controls | | | Weight | Std. Mean Difference IV, Random, 95% CI | Std. Mean Difference IV, Random, 95% CI |
|---|---|---|---|---|---|---|---|---|---|
| | Mean | SD | Total | Mean | SD | Total | | | |
| **2.1.1 Plasma** | | | | | | | | | |
| Ding YL 2017 | 563.2 | 123.4 | 20 | 40.2 | 11.2 | 20 | 4.1% | 5.85 [4.36, 7.34] | |
| Gan B 2009 | 160.54 | 32.42 | 44 | 76.18 | 20.69 | 20 | 5.7% | 2.84 [2.11, 3.57] | |
| Inomata SI 2003 | 433 | 259 | 47 | 170 | 65.9 | 25 | 6.1% | 1.22 [0.69, 1.75] | |
| Shiratori B 2014 | 150.73 | 53.43 | 37 | 74.23 | 19.34 | 30 | 6.0% | 1.81 [1.23, 2.38] | |
| Shiratori B 2017 | 59.46 | 37.7 | 36 | 40.05 | 27.72 | 19 | 6.0% | 0.55 [-0.01, 1.12] | |
| **Subtotal (95% CI)** | | | 184 | | | 114 | 27.9% | **2.28 [1.13, 3.43]** | |

Heterogeneity: Tau² = 1.55; Chi² = 58.12, df = 4 (P < 0.00001); I² = 93%
Test for overall effect: Z = 3.90 (P < 0.0001)

| | | | | | | | | | |
|---|---|---|---|---|---|---|---|---|---|
| **2.1.2 Serum** | | | | | | | | | |
| Cui JX 2014 | 588.42 | 271.67 | 80 | 169.12 | 97.43 | 100 | 6.3% | 2.14 [1.77, 2.51] | |
| Du H 2020 | 564.52 | 126.51 | 50 | 40.26 | 11.72 | 20 | 5.2% | 4.82 [3.85, 5.79] | |
| Feng PL 2017 | 691.54 | 102.72 | 55 | 171.94 | 36.83 | 60 | 5.2% | 6.81 [5.84, 7.77] | |
| Guo SX 2013 | 683.21 | 321.76 | 42 | 168.46 | 117.31 | 36 | 6.0% | 2.04 [1.49, 2.60] | |
| Hao XL 2016 | 712.62 | 335.75 | 43 | 185.34 | 120.23 | 40 | 6.1% | 2.04 [1.51, 2.58] | |
| Qiao YF 2013 | 652.4 | 270.8 | 62 | 155.6 | 107.2 | 35 | 6.1% | 2.18 [1.66, 2.70] | |
| Sun XX 2016 | 652.31 | 264.74 | 84 | 152.93 | 113.26 | 80 | 6.3% | 2.42 [2.02, 2.83] | |
| Wang Q 2019 | 663.45 | 94.2 | 92 | 384.14 | 37.15 | 50 | 6.1% | 3.51 [2.97, 4.05] | |
| Wu TF 2017 | 538.94 | 258.41 | 51 | 143.12 | 78.31 | 63 | 6.2% | 2.16 [1.69, 2.62] | |
| Wu YX 2019 | 663.9 | 282.6 | 90 | 356.4 | 130.2 | 90 | 6.4% | 1.39 [1.07, 1.72] | |
| Yuan Y 2016 | 683.2 | 321.8 | 52 | 168.5 | 117.3 | 45 | 6.1% | 2.05 [1.56, 2.55] | |
| Zhang YT 2018 | 657.5 | 315.6 | 48 | 170.3 | 114.2 | 53 | 6.1% | 2.08 [1.59, 2.57] | |
| **Subtotal (95% CI)** | | | 749 | | | 672 | 72.1% | **2.71 [2.16, 3.26]** | |

Heterogeneity: Tau² = 0.86; Chi² = 163.28, df = 11 (P < 0.00001); I² = 93%
Test for overall effect: Z = 9.68 (P < 0.00001)

| | | | | | | | | | |
|---|---|---|---|---|---|---|---|---|---|
| **Total (95% CI)** | | | 933 | | | 786 | 100.0% | **2.58 [2.09, 3.08]** | |

Heterogeneity: Tau² = 0.98; Chi² = 236.86, df = 16 (P < 0.00001); I² = 93%
Test for overall effect: Z = 10.21 (P < 0.00001)
Test for subgroup differences: Chi² = 0.44, df = 1 (P = 0.51), I² = 0%

**Fig 2. The result of association between serum/plasma OPN levels and tuberculosis.**

| Study or Subgroup | AFB(+) | | | AFB(-) | | | Weight | Std. Mean Difference IV, Random, 95% CI | Std. Mean Difference IV, Random, 95% CI |
|---|---|---|---|---|---|---|---|---|---|
| | Mean | SD | Total | Mean | SD | Total | | | |
| Feng PL 2017 | 637.28 | 125.31 | 35 | 435.72 | 100.65 | 20 | 13.9% | 1.70 [1.06, 2.34] | |
| Guo SX 2013 | 612.48 | 340.33 | 24 | 426.24 | 154.31 | 18 | 14.2% | 0.66 [0.03, 1.29] | |
| Wang Q 2019 | 714.52 | 102.63 | 48 | 607.74 | 86.73 | 44 | 20.0% | 1.11 [0.67, 1.55] | |
| Wu YX 2019 | 741.2 | 257.8 | 41 | 599.5 | 219 | 49 | 20.6% | 0.59 [0.17, 1.02] | |
| Yuan Y 2016 | 612.5 | 340.3 | 30 | 426.2 | 154.3 | 22 | 16.0% | 0.66 [0.09, 1.23] | |
| Zhang YT 2018 | 745.3 | 304.4 | 27 | 544.6 | 183.1 | 21 | 15.2% | 0.76 [0.17, 1.36] | |
| **Total (95% CI)** | | | 205 | | | 174 | 100.0% | **0.90 [0.58, 1.21]** | |

Heterogeneity: Tau² = 0.08; Chi² = 10.28, df = 5 (P = 0.07); I² = 51%
Test for overall effect: Z = 5.58 (P < 0.00001)

**Fig 3. The result of association between serum/plasma OPN levels and AFB staining of sputum.**

integrins and CD44 variants at the cell surface, mediating cell adhesion, migration and survival in a variety of inflammatory cells including T cells, macrophages and NK cells [33, 34]. Of interest, OPN was recently characterized as cleavage sites by several proteases including thrombin, plasmin and matrix metalloproteinases (MMPs) [35]. In many cases, the cleaved forms of OPN demonstrated augmented cell bindings, inducing enhanced adhesion and

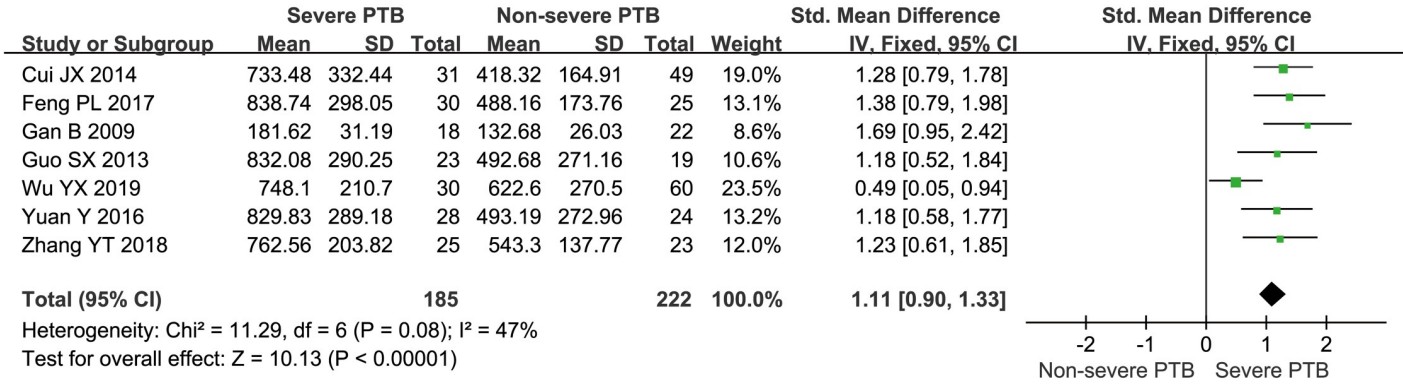

**Fig 4. The result of association between serum/plasma OPN levels and imaging severity of pulmonary tuberculosis.**

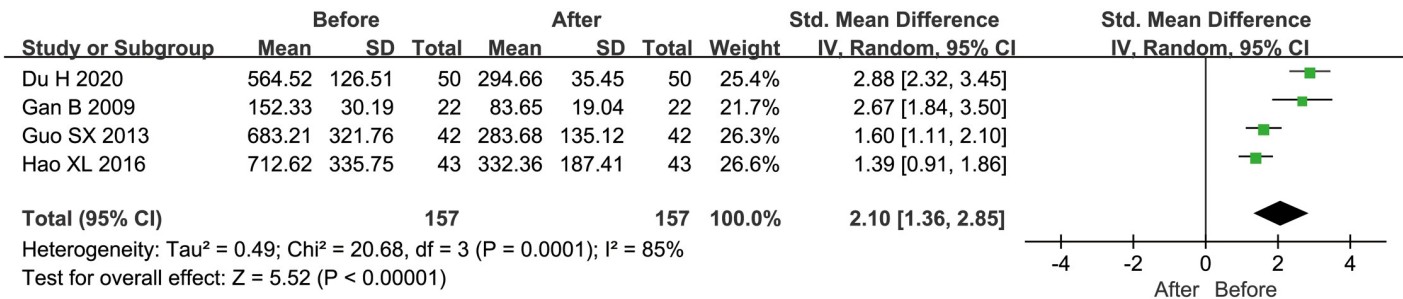

**Fig 5. The result of association between serum/plasma OPN levels and anti-tuberculous therapy.**

migration in vitro, which perhaps caused by conformational changes of OPN fragments, however, the functional role of OPN cleavage in vivo is largely unexplored [36]. Since the initial identification for OPN as one of the most highly up-regulated genes upon T cell activation, it has been demonstrated multiple roles in regulating host immune response in the last decades [37]. By the transgenic mice model, soluble OPN has been observed an induction to proliferation of effector CD4[+] and CD8[+] cells in cell-mediated reactions, while OPN[-/-] mice demonstrated severely impaired cell-mediated immunity to viral and bacterial infections [38, 39]. Here, we reviewed the potential of OPN for diagnosis, severity assessment and therapeutic effect monitoring of TB.

Consistent with the previous studies, our results further strengthened the evidence linking OPN expressions with MTB pathogenicity. Shiratori and colleagues identified that OPN had the discriminatory capacity to tuberculosis with sensitivity of 94.6% and specificity of 93.3% [13]. In addition, thorough analyses for the relationship of OPN secretions to sputum AFB staining and imaging severity of infections also achieved useful conclusions that patients with positive AFB results and severe infections got higher levels of OPN in blood. Sevtekin et al. examined the OPN levels in cattle tuberculosis and observed a marked increase of OPN concentrations in tuberculous lesions, whereas no OPN expression was detected in normal tissues by immunohistochemistry [40]. Moreover, higher OPN expressions were detected in patients with active and latent TB infections compared to healthy controls [8], and circulating IFN-γ and OPN paralleled to the extent of lung lesions [15]. Cell-mediated adaptive immunity is crucial for host defense to *M. Tuberculosis*, while little is known about complete interactions between cytokines and immune cell behaviors during

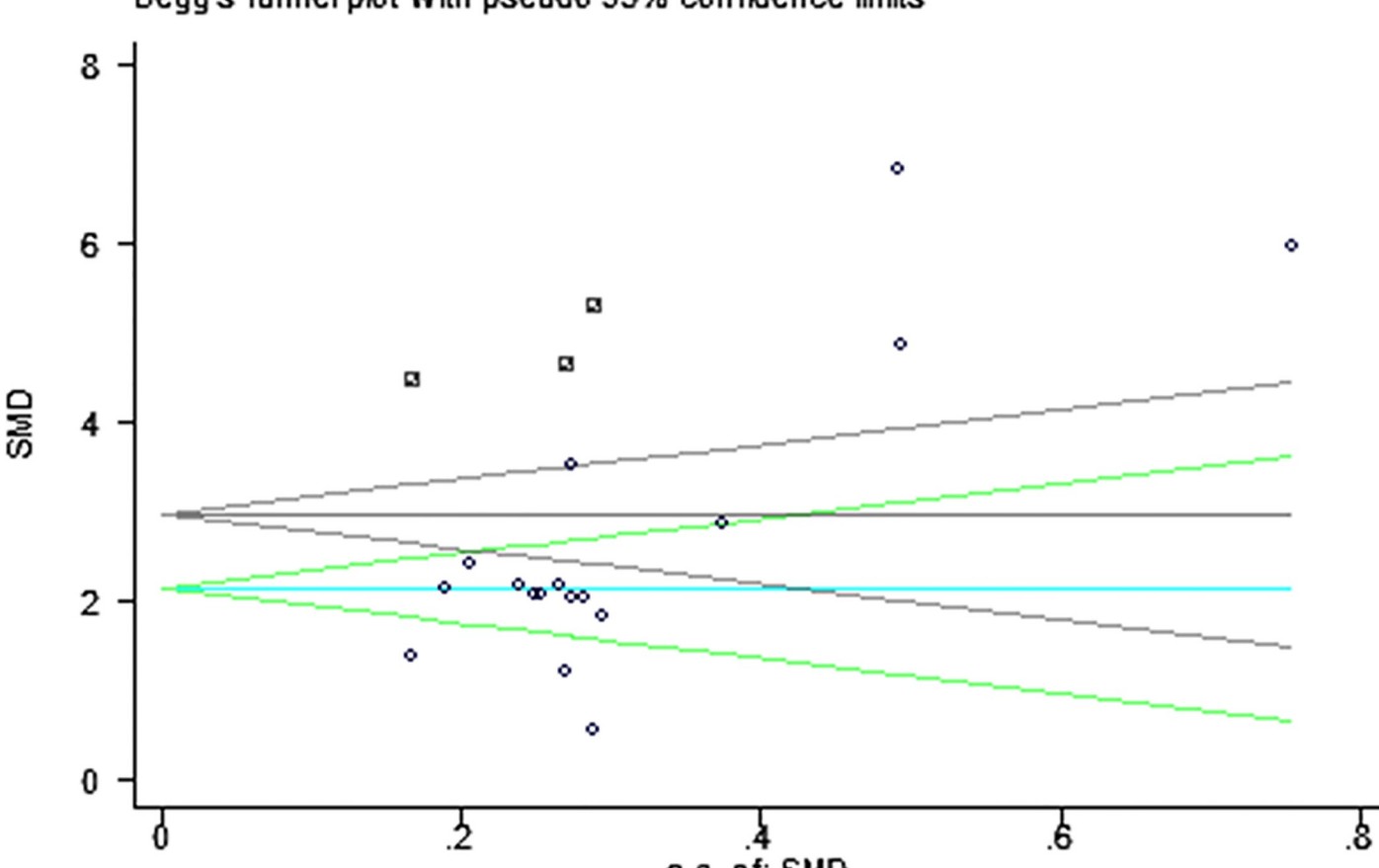

**Fig 6. The Begg's test and trim-and-fill analysis.**

this process, and studies from OPN gene-deficient models will provide us further insights into the pathophysiological role of OPN. Compared with wild type mice, OPN$^{-/-}$ mice had more severe *M. Tuberculosis* infections characterized by heavier loads and delayed clearance of bacteria, and *M. Tuberculosis* grew more rapidly in OPN-null macrophages [13, 41, 42]. Several studies demonstrated that OPN regulated macrophages and T cells migration, activation and cytokine expression in tuberculosis, mediating accumulations of macrophages, macrophage-derived epithelioid cells, and giant cells during the granuloma formation, possibly explained the impaired ability of OPN-deficient hosts to control mycobacterial infection [43, 44]. Furthermore, deficient OPN gene expression inhibited IFN-γ production as well, contributing to severely impaired cell-mediated immunity and granulomas maturity in mice [39]. To sum up, a suggested explanation is that OPN is highly expressed by activated immune cells during the MTB infections, in addition, it also co-stimulates T cell proliferation in the setting of T cell activation and induces the expressions of Th1 but not Th2 cytokines like IL-2, IFN-γ and TNF-α in macrophages and T cells, helping to recruit immune cells to the site of tuberculous lesions and mediate phagocytosis of macrophages and granuloma formation [13, 42]. However in Van Der Windt's study [9], similar bacterial burden, tissue inflammation and recruitment of immune cells were observed in lungs from both wild type and OPN knocked out mice after 2 and 5 weeks of *M. Tuberculosis* infection, despite the OPN expression was up-regulated in alveolar

macrophages and lymphocytes, demonstrating that OPN might not be crucial for the protection upon *M. Tuberculosis* infection. While the intracellular signaling pathways activated by OPN and roles of cleaved OPN fragments in MTB pathogenesis have not been known precisely by now, more studies are needed in the future.

With regard to OPN and *M. Tuberculosis* infection, previous studies have revealed a significant fall of plasma OPN levels after anti-tuberculous chemotherapy [45, 46], and our results also supported this notion by demonstrating that OPN concentrations decreased paralleling with the clinical improvement after treatment. In vitro studies by peripheral blood mononuclear cells infected with *M. Tuberculosis* found an apparently reduced production of IFN-γ and IL-12 with neutralizing anti-OPN monoclonal antibody [45], suggesting circulating OPN may serving as a reliable indicator of improvement during the early stage of anti-tuberculous treatment regimens.

Despite the divergence of serum/plasma OPN potencies, OPN was also detected significantly differential expressions in sputum and pleural effusions with various aetiologies by several studies. Tian and colleagues found that OPN levels in sputum supernatant was obviously higher than those in healthy controls [47]. In another study prospectively investigating the OPN concentrations in pleural effusions of different aetiologies, researchers demonstrated that OPN levels were significantly elevated in exudative pleural effusions compared to those in blood or transudative effusions, and the higher pleural effusion/serum OPN ratio was observed in malignancies than that in tuberculous effusions [48], which could be useful for diagnostic purposes.

Our systematic review and meta-analysis have several limitations that should be noted. Firstly, wide heterogeneity was observed in this study. Although we conducted a meta-regression on the factors may causing heterogeneity such as gender, average age, sample size, tuberculosis burden, immune status and lesion sites, possible sources of heterogeneity were not seeked out statistically. Currently, the validated diagnostics for active tuberculosis mainly include microscopy, cultures and nucleic acid amplification tests (NAATs) such as Xpert MTB/RIF and loop-mediated amplification test (LAMP). For screening of tuberculosis, imaging with digital radiology and computer-aided interpretations becomes a widely used method [49]. The diversity of diagnostic methods may cause the heterogeneity among studies as well. In this meta-analysis, the detection methods of tuberculosis contained AFB staining, culture, histopathology, etc., and several studies didn't report the detailed inclusion criteria for tuberculosis patients, among which patients with experimental anti-tuberculosis treatment might exist. Hence, there is reason to believe that the diagnostic method is one of the origins of high heterogeneity. Secondly, unpublished or other ongoing trials were not retrieved, and even among the included studies, most of which were from high disease burden countries, and as a result, the possibility of publication bias was not excluded exactly. Thirdly, the nature of retrospective study design leads to the identification of association but not causality link. All involved studies were case-control or cross-sectional, which simply meant the relevance between abnormal OPN levels and tuberculosis, while it is still not clear whether elevated OPN levels are the contributor to tuberculosis or not. Fourthly, depending on the alternative splicing, the OPN has now several isoforms called the full-length variant and cleaved variants, all of which may display distinct functions. Among the included studies, only two of them reported the forms of OPN they detected. While Boggio and colleagues recently reported a similar activation of IFN-γ secretion in T cells by full-length and cleaved OPN stimulation, indicating that different OPN variants might be similarly active in anti-TB immune response [50]. And lastly, limited to the characteristics of included studies we couldn't scientifically evaluate the diagnostic

efficiency of OPN to tuberculosis and differential value to other diseases. Thereby, further prospective, longitudinal and well-designed cohort studies are needed.

## Conclusions

In this comprehensive systematic review and meta-analysis, we found that elevated serum/plasma OPN concentrations were associated with an increased risk of tuberculosis especially positive smear tuberculosis in retrospective studies. What's more, higher OPN expressions were related to imaging-severe tuberculosis, and OPN levels in tuberculosis patients decreased after efficient anti-tuberculosis therapies. The results provided an improved understanding of OPN as a potential biomarker for tuberculosis diagnosis, evaluation and therapeutic monitering. Undoubtedly, further prospective, large and well-designed cohort studies are needed to elucidate the exact role of OPN in tuberculosis development.

## Supporting information

**S1 Checklist. PRISMA 2009 checklist.**
(DOC)

**S1 Table. Meta-regression analysis for potential sources of heterogeneity.**
(DOCX)

**S1 Fig. The result of sensitivity analysis on association between serum/plasma OPN levels and tuberculosis.**
(TIF)

**S2 Fig. The funnel plot of publication bias.**
(TIF)

## Author Contributions

**Data curation:** Dongguang Wang, Lian Wang, Shijie Zhang, Jizhen Huang, Li Zhang.

**Formal analysis:** Dongguang Wang, Xiang Tong, Lian Wang.

**Funding acquisition:** Hong Fan.

**Methodology:** Dongguang Wang.

**Project administration:** Hong Fan.

**Resources:** Dongguang Wang.

**Supervision:** Hong Fan.

**Validation:** Hong Fan.

**Writing – original draft:** Dongguang Wang, Xiang Tong.

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
