## [Decision Letter · Decision Letter 0]

20 Aug 2020

PONE-D-20-17846

The association between osteopontin and tuberculosis risk, severity and prognosis: a systematic review and meta-analysis

PLOS ONE

Dear Dr. Fan,

Thank you for submitting your manuscript to PLOS ONE. After careful consideration, we feel that it has merit but does not fully meet PLOS ONE’s publication criteria as it currently stands. Therefore, we invite you to submit a revised version of the manuscript that addresses the points raised during the review process.

We look forward to receiving your revised manuscript.

Kind regards,

Martin E Rottenberg

Academic Editor

PLOS ONE

Journal Requirements:

2. Please include captions for your Supporting Information files at the end of your manuscript, and update any in-text citations to match accordingly. Please see our Supporting Information guidelines for more information: http://journals.plos.org/plosone/s/supporting-information

Reviewers' comments:

Reviewer's Responses to Questions

**Comments to the Author**

1. Is the manuscript technically sound, and do the data support the conclusions?

Reviewer #1: Yes

Reviewer #2: Partly

Reviewer #3: Yes

2. Has the statistical analysis been performed appropriately and rigorously? 

Reviewer #1: I Don't Know

Reviewer #2: I Don't Know

Reviewer #3: Yes

3. Have the authors made all data underlying the findings in their manuscript fully available?

Reviewer #1: No

Reviewer #2: Yes

Reviewer #3: Yes

4. Is the manuscript presented in an intelligible fashion and written in standard English?

Reviewer #1: Yes

Reviewer #2: No

Reviewer #3: No

5. Review Comments to the Author

Reviewer #1: The paper by Dr. Wang reported the systemic review regarding the association between osteopontin and tuberculosis. This is the new analysis but the results only supported the previous and did not get any new findings from meta analysis. They did not analyze detailed methods for the analysis. Comparison of serum of plasma should be done. Dose elisa could detect full length OPN or cleaved form of OPN ? Finally which biological functions of OPN are associated with tuberculosis. It should be described which control diseases were used for each data analysis.

Reviewer #2: This paper is a meta-analysis of the use of serum / plasma osteopontin in tuberculosis. The authors present pooled anaysis for four parameters:

1. OPN is higher in tuberculosis than controls (healthy controls)

2. OPN is higher in smear positive than smear negative TB

3. OPN is higher in severe TB than non-severe.

4. OPN levels fall in response to treatment.

As regards the first point, the paper does not add anything new. We can not evaluate sensitivity, specificity etc from the pooled data and elevated levels in TB vs healthy controls has been well documented in the quoted publications. Analysis of parameters 2-4 are nicely presented and clarify these issues.

The authors state that their analysis provides data on risk, severity and prognosis but the data ony concerns diagnosis and severity. This should be restated in the paper.

The authors should address the issue of OPN as a diagnostic tool to differentiate TB vs other disease, in addition to their data vs healthy controls.

Have the authors checked that the quoted articles to exclude double entries of the same subjects from different publication?

English - there are "google translate" mistakes that need correction such as "affluently" (p2), acid "fasting" (p4), inherent "vice "of macrophages (p5), "remarked" accumulation (p5)

Reviewer #3: This paper reports on a meta-analysis of tuberculosis. As it stands the data presented compare cases and controls in terms of their osteopontin levels. The title really needs to look at this - the prognosis is not really presented as this would require a cohort study and not a case control study.

The analysis of severe vs mild here eliminates some data (i.e. the controls) and the analysis here needs to look at the severity on the whole.

Please explain whether proper meta-analytic methods were used or simply pooled analyses - pooled analyses and meta-analyses are not the same.

Please provide the Q statistic as this is an important thing in the weighting of the studies.

Please do not use p=0.000 as p is notexactly zero.

Why are studies ordered as they are in the plots? The units on the axis are not particularly informative.

6. PLOS authors have the option to publish the peer review history of their article (what does this mean?). If published, this will include your full peer review and any attached files.

Reviewer #1: No

Reviewer #2: No

Reviewer #3: No

---

## [Author Response · Author response to Decision Letter 0]

26 Sep 2020

Dear Editor-in-chief and Reviewers,

We would like to thank you for appreciating our work, and also we’d like to thank the reviewers for making many thoughtful comments.

According to the comments from reviewers, we have revised the manuscript entitled “The association between osteopontin and tuberculosis risk, severity and prognosis: a systematic review and meta-analysis (Manuscript No.: PONE-D-20-17846)”. We revised our manuscript by the PLOS ONE’s style requirements and we are looking forward to the article can be published in your journal.

Here are our point-by-point responses:

Reviewer #1

The paper by Dr. Wang reported the systemic review regarding the association between osteopontin and tuberculosis. This is the new analysis but the results only supported the previous and did not get any new findings from meta analysis.

Author response:

Thank you for your time and constructive comments. All suggested corrections have been addressed as follows:

1. They did not analyze detailed methods for the analysis. Comparison of serum of plasma should be done.

Author response:

We followed the reviewer’s suggestions and performed the subgroup analysis by plasma and serum, and the result showed no change to heterogeneity (lines 58-60).

2. Dose elisa could detect full length OPN or cleaved form of OPN?

Author response:

Thank you for your remark. In the present meta-analysis, only two of the seventeen included studies reported the forms of detected OPN variants (line 169, Table 2). OPN has two terminal zones including N-terminal and C-terminal, and C-terminal binds two heparin molecules as well as CD44 variants whereas N-terminal includes integrin receptor binding zones. Although diverse sites have been reported, the functional significance of cleaved fragment of OPN remains unknown. In many cases, the cleaved forms of OPN demonstrated augmented cell bindings, inducing enhanced adhesion and migration in vitro (lines 229-231). While in another study, researchers reported a similar activation of IFN-γ secretion in T cells by full-length and cleaved OPN stimulation, indicating that different OPN variants might be similarly active in anti-TB immune response (Lines 336-339). Based on that, further well-designed studies will be needed to reveal the precise role of cleaved OPN in the pathogenesis of tuberculosis in the future. In the revised version, we added this statement to table, result, discussion and limitation sections to make readers better understand the role of different OPN variants.

3. Finally which biological functions of OPN are associated with tuberculosis.

Author response:

Thanks for your valued query. As we know, cell-mediated immune responses play a critical role in host defense to MTB infections. OPN regulates macrophages and T cells migration, activation and cytokine expression in tuberculosis. While hosts infect with MTB, activated lymphocytes and other immune cells highly express OPN, which is a significant chemical attractant for macrophages and T cells. OPN supports adhesion and induces migration of T cells and macrophages, and also, it co-stimulates T-cell proliferation and induces T cells and macrophages expressing Th1 cytokines like IFN-γ and IL-12, which help to recruit immune cells to the site of tuberculous lesions and mediate phagocytosis of macrophages and granuloma formation (lines 271-277). However, the intracellular signaling pathways activated by OPN have not been known precisely by now, more studies are needed in the future (lines 283-286). In the revised version, we added this statement to discussion section to make readers better understand the role of OPN.

4. It should be described which control diseases were used for each data analysis.

Author response:

Thank you for raising this point. To make the control and experimental groups in each data analysis more intuitive, we displayed the results in forest plots by RevMan 5.2, and at the same time, the pooled results were changed and we corrected them in the revised manuscript.

Reviewer #2

1. As regards the first point, the paper does not add anything new. We can not evaluate sensitivity, specificity etc from the pooled data and elevated levels in TB vs healthy controls has been well documented in the quoted publications. Analysis of parameters 2-4 are nicely presented and clarify these issues.

Author response:

Thank you for your time and constructive comments. In the present article, few included studies evaluate the sensitivity, specificity, etc. of OPN on tuberculosis diagnosis, severity and therapeutic monitoring, and the included populations in most original studies are tuberculosis patients and healthy controls, therefore, we couldn’t calculate sensitivity, specificity, etc. from the existing data, and also we stated the limitations in discussion section. Additionally, we listed the limited data of original text in discussion section as a literature review of the relationship between OPN and tuberculosis.

2. The authors state that their analysis provides data on risk, severity and prognosis but the data only concerns diagnosis and severity. This should be restated in the paper.

Author response:

Thanks for your valued remark. We reported the role of OPN on tuberculosis diagnosis, imaging severity evaluation and therapeutic effect monitoring in this meta-analysis, and therefore, we replaced the title by “The association between osteopontin and tuberculosis: a systematic review and meta-analysis” and corrected the expressions in main text of the revised version to avoid misleading readers.

3. The authors should address the issue of OPN as a diagnostic tool to differentiate TB vs other disease, in addition to their data vs healthy controls.

Author response:

Thank you for raising this point. According to the reviewer’s recommendation, we retrieved studies on OPN as a diagnostic tool to differentiate TB and other diseases again, and unfortunately the results were limited and we couldn’t catch a pooled result on the differential diagnosis of OPN to other diseases in this meta-analysis. Instead, we reviewed the limited studies comparing tuberculosis and other diseases (pleural effusions with different aetiologies) and summarized their findings in the revised manuscript to better understand the value of OPN in clinical research in the future (lines 297-307).

4. Have the authors checked that the quoted articles to exclude double entries of the same subjects from different publication?

Author response:

Thank you for your query. We checked and excluded the repeated studies from different publications at the stage of study selection and exclusion.

5. English - there are "google translate" mistakes that need correction such as "affluently" (p2), acid "fasting" (p4), inherent "vice "of macrophages (p5), "remarked" accumulation (p5)

Author response:

Thank you. According to the reviewer’s recommendation, we have corrected the misnomers in revised version.,

Reviewer #3

This paper reports on a meta-analysis of tuberculosis. As it stands the data presented compare cases and controls in terms of their osteopontin levels.

Author response:

Thank you for your time and all suggested corrections have been addressed as follows:

1. The title really needs to look at this - the prognosis is not really presented as this would require a cohort study and not a case control study.

Author response:

Thanks for your valued query. We reported the role of OPN on tuberculosis diagnosis, imaging severity evaluation and therapeutic effect monitoring in this meta-analysis, and therefore, we replaced the title by “The association between osteopontin and tuberculosis: a systematic review and meta-analysis” and corrected the expressions in main text of the revised version to avoid misleading readers.

2. The analysis of severe vs mild here eliminates some data (i.e. the controls) and the analysis here needs to look at the severity on the whole.

Author response:

Thank you for your query. The pooled data showed that serum/plasma concentrations of OPN in tuberculosis patients were higher than those in healthy individuals. Furthermore, according to the imaging findings from included studies, the pulmonary tuberculosis patients could be divided into two groups: severe PTB (including military tuberculosis and cavitary tuberculosis) and non-severe PTB (including infiltrative tuberculosis and tuberculous pleurisy). We compared the OPN levels between two groups based on imaging severity, and we have corrected the inaccurate statement in the revised manuscript (lines 176-184).

3. Please explain whether proper meta-analytic methods were used or simply pooled analyses - pooled analyses and meta-analyses are not the same.

Author response:

Thanks. We consulted a professor of statistics in conducting this meta-analysis, therefore, proper meta-analytic methods were used in this study.

4. Please provide the Q statistic as this is an important thing in the weighting of the studies.

Author response:

Thank you for the remark. We evaluated the heterogeneity by RevMan 5.2 and the heterogeneity test results (Q statistic, P value and I2) were illustrated in the revised figures (Fig2-5).

5. Please do not use p=0.000 as p is notexactly zero.

Author response:

Thank you. The P values in the original figures are generated by STATA software automatically and the corrected P values were shown as P＜0.00001 by RevMan 5.2 in Fig 2-5.

6. Why are studies ordered as they are in the plots? The units on the axis are not particularly informative.

Author response:

Thanks for the remark. We validated the results of the meta-analysis by RevMan 5.2 and enriched the presentations in the updated figures to make them more intuitive for readers.

Thank you for your time and efforts.

Best regards!

Hong Fan

---

## [Decision Letter · Decision Letter 1]

9 Nov 2020

The association between osteopontin and tuberculosis: a systematic review and meta-analysis

PONE-D-20-17846R1

Dear Dr. Fan,

We’re pleased to inform you that your manuscript has been judged scientifically suitable for publication and will be formally accepted for publication once it meets all outstanding technical requirements.

Kind regards,

Martin E Rottenberg

Academic Editor

PLOS ONE

Additional Editor Comments (optional):

Reviewers' comments:

Reviewer's Responses to Questions

**Comments to the Author**

1. If the authors have adequately addressed your comments raised in a previous round of review and you feel that this manuscript is now acceptable for publication, you may indicate that here to bypass the “Comments to the Author” section, enter your conflict of interest statement in the “Confidential to Editor” section, and submit your "Accept" recommendation.

Reviewer #1: All comments have been addressed

Reviewer #3: All comments have been addressed

2. Is the manuscript technically sound, and do the data support the conclusions?

Reviewer #1: Yes

Reviewer #3: (No Response)

3. Has the statistical analysis been performed appropriately and rigorously? 

Reviewer #1: Yes

Reviewer #3: (No Response)

4. Have the authors made all data underlying the findings in their manuscript fully available?

Reviewer #1: Yes

Reviewer #3: (No Response)

5. Is the manuscript presented in an intelligible fashion and written in standard English?

Reviewer #1: Yes

Reviewer #3: (No Response)

6. Review Comments to the Author

Reviewer #1: The paper described an interesting subject; OPN in tuberculosis. Though the manuscript dose not give new aspects of OPN in MTB. It is valuable to summarize data for publication.

Reviewer #3: (No Response)

7. PLOS authors have the option to publish the peer review history of their article (what does this mean?). If published, this will include your full peer review and any attached files.

Reviewer #1: **Yes: **Toshio Hattori

Reviewer #3: No

---

## [Editor Report · Acceptance letter]

18 Nov 2020

PONE-D-20-17846R1 

The association between osteopontin and tuberculosis: a systematic review and meta-analysis 

Dear Dr. Fan:

I'm pleased to inform you that your manuscript has been deemed suitable for publication in PLOS ONE. Congratulations! Your manuscript is now with our production department. 

Kind regards, 

on behalf of

Dr. Martin E Rottenberg 

Academic Editor

PLOS ONE